# Epitope Mapping of the Diphtheria Toxin and Development of an ELISA-Specific Diagnostic Assay

**DOI:** 10.3390/vaccines9040313

**Published:** 2021-03-26

**Authors:** Salvatore Giovanni De-Simone, Larissa Rodrigues Gomes, Paloma Napoleão-Pêgo, Guilherme Curty Lechuga, Jorge Soares de Pina, Flavio Rocha da Silva

**Affiliations:** 1Center for Technological Development in Health (CDTS), Oswaldo Cruz Foundation (FIOCRUZ), National Institute of Science and Technology for Innovation in Neglected Diseases Populations (INCT-IDNP), Rio de Janeiro 21040-900, Brazil; larissa.gomesr@cdts.fiocruz.br (L.R.G.); paloma.pego@cdts.fiocruz.br (P.N.-P.); guilherme.lechuga@cdts.fiocruz.br (G.C.L.); jorge.pina@cdts.fiocruz.br (J.S.d.P.); flavio.rocha@cdts.fiocruz.br (F.R.d.S.); 2Molecular and Cellular Biology Department, Biology Institute, Federal Fluminense University, Niterói 24020-141, Brazil

**Keywords:** diphtheria toxin, B epitopes, epitope mapping, bi-specific peptides, ELISA, diagnostic performance

## Abstract

*Background:* The diphtheria toxoid antigen is a major component in pediatric and booster combination vaccines and is known to raise a protective humoral immune response upon vaccination. Although antibodies are considered critical for diphtheria protection, little is known about the antigenic determinants that maintain humoral immunity. *Methods:* One-hundred and twelve 15 mer peptides covering the entire sequence of diphtheria toxin (DTx) protein were prepared by SPOT synthesis. The immunoreactivity of membrane-bound peptides with sera from mice immunized with a triple DTP vaccine allowed mapping of continuous B-cell epitopes, topological studies, multiantigen peptide (MAP) synthesis, and Enzyme-Linked Immunosorbent Assay (ELISA) development. *Results:* Twenty epitopes were identified, with two being in the signal peptide, five in the catalytic domain (CD), seven in the HBFT domain, and five in the receptor-binding domain (RBD). Two 17 mer (CB/Tx-2/12 and CB/DTx-4–13) derived biepitope peptides linked by a Gly-Gly spacer were chemically synthesized. The peptides were used as antigens to coat ELISA plates and assayed with human (huVS) and mice vaccinated sera (miVS) for in vitro diagnosis of diphtheria. The assay proved to be highly sensitive (99.96%) and specific (100%) for huVS and miVS and, when compared with a commercial ELISA test, demonstrated a high performance. *Conclusions:* Our work displayed the complete picture of the linear B cell IgG response epitope of the DTx responsible for the protective effect and demonstrated sufficient specificity and eligibility for phase IIB studies of some epitopes to develop new and fast diagnostic assays.

## 1. Introduction

Diphtheria is a vaccine-preventable, bacterial disease caused by toxin-producing strains of *Corynebacteria diphtheriae* and is once again a growing concern. It is endemic in Asia (India, Indonesia, Iran, Nepal, Pakistan, etc.), Africa (Ghana), and some countries of South America [1,2,3] but well-controlled in countries with high vaccination coverage. However, in the last few years it has been re-emergent in parts of Europe [4] and Venezuela [5], and there have been imported cases in other countries of Europe [6,7,8,9], the United States of America [10], and Brazil [5].

Not only *C. diphtheriae* but also *Corynebacterium ulcerans* and *Corynebacterium pseudotuberculosis* have the potential to produce diphtheria toxin (DTx) and hence can cause classic respiratory diphtheria (spread by droplets) as well as cutaneous diphtheria (direct contact) [11].

Due to the potential local or systemic spread of DTx, classical diphtheria may give rise to severe life-threatening respiratory symptoms characterized by the development of an adherent pseudomembrane in the upper respiratory tract as well as myocarditis and polyneuritis with a fatality rate between 5 and 30% [6]. In contrast, cutaneous diphtheria is typically characterized by well-demarcated ulcers that might have membranes. The lesions are slow healing and might act as a reservoir from which bacteria can be transmitted to susceptible contacts, potentially resulting in cutaneous or respiratory disease [12].

Disease severity is mediated by successful bacterial secretion of the potent DTx that inhibits protein synthesis in eukaryotic cells by disrupting elongation factor 2 (EF2) function, causing cell death [13]. DTx consists of a single protein with a disulfide bond linking two fragments processed by trypsin [14,15]. Fragment A contains the catalytic domain (aa1–aa188) [15], and fragment B the translocation (PFMT, aa200–aa378) and receptor-binding domains (RBD, aa387–aa535) [16].

Vaccination with diphtheria toxoid-containing vaccine might not prevent cutaneous colonization or infection with *C. diphtheriae* [2,12]. Non-toxin-producing strains of *C. diphtheriae* can also cause disease that is generally less severe, although attack associated with non-toxin-producing strains has been reported [17]. As the mortality of infections caused by the three corynebacteria species mentioned above is almost entirely due to DTx [17,18,19,20,21,22,23,24,25,26,27,28], the availability of rapid methods for the identification of the species as well as for the detection of DTx or the DT tox gene is of primary importance.

DTxs produced by different corynebacteria are homologues [29,30,31,32] and the identification and characterization of corynebacteria are conducted by phenotypic methods [11,33,34], polymerase chain reaction (PCR) [35,36,37], and mass spectrometry (MALDI-TOFF) [38]. The immunological tests developed for the disease diagnostic were the microcell culture [39], fluorescence immunoassay [40], and duple antigen [41] or sandwich-Enzyme-Linked Immunosorbent Assay (ELISA) [42].

The diphtheria toxoid antigen is a major component in pediatric and booster combination vaccines and is known to raise a protective humoral immune response upon vaccination. However, a structurally resolved analysis of TxD epitopes with underlying molecular mechanisms of antibody neutralization is scarce or has not yet been revealed.

Two neutralizing anti-DTx monoclonal antibodies (mAbs) recognizing conformational epitopes blocking the heparin-binding epidermal grow factor (HBEGF) binding site have been identified by mass spectrometry/interferometry and used to develop an ELISA [43]. Other approaches used for identifying epitopes/blocking antibodies are the phage display assay [44,45], targeting of the specific domain [46], and human mAbs isolated from vaccinated volunteers [47]; however, most of the epitopes were not identified or long peptides sequences have been proposed. A recent study, using dynamic simulation and temperature variation, also predicted some conformation of B-cell epitopes in DTX [48].

In this work, using a more refined methodology of epitope synthesis, we report the mapping of B-cell continuous epitopes of DTx, the topological characterization of the epitopes, and the production by chemical synthesis of DTx-bi-epitope and its use as antigen for diphtheria diagnosis. The printable SPOT synthesis is an accessible and very stretchy technique for microarray production of peptides on membrane supports. Furthermore, it allows rapid and low-cost access to a large number of peptides for systematic epitope analysis [49]. The biepitope MAP4 used as coating antigen in ELISA accurately distinguishes high sensitivity and specificity sera of immunized mice and humans, from healthy animals and individuals.

## 2. Materials and Methods

### 2.1. Materials and Antibodies

Amino acids, reagents for peptide synthesis, casein, goat antihuman IgG-biotin, rabbit antimouse IgG-biotin, peroxidase labeled-neutravidin and 3,3′,5, 5’tetramethyl benzidine (TMB) were from Sigma-Merck (St Louis, MO, USA). Cellulose membranes of the ester type were from Intavis (Koeln, Germany). Alkaline phosphatase labeled sheep antimouse IgG (H+L) was obtained from KPL (Sherman Oaks, CA, USA). HRP labeled neutravidin, and SuperSignal West Pico chemiluminescent substrate was from Thermo (Waltham, MA, USA). Immulon 2HB flat-bottom 96-well microtiter plates were from Corning. All other reagents were of analytical grade. The triple DTP vaccine used during the vaccination of children/infants containing inactivated whole-cell bacterium of *B. pertussis* combined with formaldehyde-inactivated diphtheria and tetanus toxoid was produced by Bio-Manguinhos (FIOCRUZ, RJ, Brazil).

### 2.2. Human, Horse, and Mice Sera

Ninety-two children, aged 6–16 years (median age 7.5 years), vaccinated with the whole DTP with no evidence of acute infection or known history of whooping cough and diphtheria were enrolled in this study. One hundred sera samples from healthy blood bank donators (HEMORIO) were included in the study.

To obtain mice sera, thirty-eight NIH Swiss mice (12–16 g) were immunized as described previously [50], with the DTP (diphteriae/tetanus/pertussis) vaccine reconstituted with saline and 2 IU (defined by the Brazilian National Immunization Program) administered at 0.5 mL with an interval of 21 days. Sera were collected one week after the last inoculation and stored at −20 °C.

The commercial horse therapeutic (hoThe) sera was obtained from Butantan Institute (Lot 170166), São Paulo, SP, Brazil.

The procedures involving the animals and their care were conducted following the Guidelines for the Use of Animals in Biochemical Research/FIOCRUZ (INCQs-CEUA P0137/02). The study was approved by the UNIGRANRIO (CAAE: 24856610.0.0000.5283) study center ethics committee and conducted under good clinical practice and all applicable regulatory requirements including the Declaration of Helsinki.

### 2.3. Synthesis of the Cellulose Membrane-Bound Peptide Array

The entire sequence of the DTx (Q6NK15, strain ATCC 700971/NCTC 13129/Biotype gravis, 560 aa) was covered with the synthesis of 15-residue-long peptides with overlapping of 10 residues, automatically prepared on cellulose membranes according to standard SPOT synthesis protocol using an Auto-Spot Robot ASP-222 (Intavis Bioanalytical Instruments AG, Köln, Germany).

The GSGSG sequence was used as a spacer at the beginning (spot A1) and end (spot E16) of the protein sequence and the programming was carried out with the Multipep software (Intavis Bioanalystical Instruments AG, Köln, Germany). Negative [QEVRKYFCV, Vaccinia virus spot F9 and F17] and positive [KEVPALTAVETGATN (Poliovirus, spots F3 and F11)/GYPKDGNAFNNDRI (*Clostridium tetani*, spot F5 and F13)/YDYDVPDYAGYP YDV (*H. influenza* virus hemagglutinin, spot F7, and F15)] controls were included. The entire library contained 112 peptides plus 6 positive and 2 negative control peptides.

The coupling reactions were followed by acetylation with acetic anhydride (4%, *v*/*v*) in N, N-dimethylformamide to render the peptides N-reactive during subsequent steps. After acetylation, the F-moc protecting groups were removed by the addition of piperidine to make the nascent peptides reactive. The remaining amino acids were added by this same coupling, blocking, and deprotection process until the desired peptide was generated. After the addition of the last amino acid, the side chains of the amino acids were deprotected using a solution of dichloromethane-trifluoracetic acid-triisopropyl silane (1: 1: 0.05, *v*/*v*/*v*) and washed with ethanol as described previously [51]. Membranes containing the synthetic peptides were probed immediately.

### 2.4. Screening of SPOT Membranes

SPOT membranes were washed for 10 min with TBS-T (50 mM Tris,136 mM NaCl, 2 mM KCl and 0.05 Tween, pH 7.4) and then blocked with TBS-T (containing 1.5% BSA) for 90 min at 8 °C under agitation. After extensive washing with TBS-T, membranes were incubated for 12 h with a pool (*n* = 30) of mice vaccinated sera diluted (1:150 for IgG detection) in TBS-T + 0.75% BSA and then washed again with TBS-T. After that, membranes were incubated with sheep antimouse IgG alkaline phosphatase labeled (diluted 1:5000), prepared in TBS-T + 0.75% BSA for 1 h, and then washed with TBS-T and CBS (50 mM citrate-buffer saline). The chemiluminescent substrate Nitro-Block II was added to complete the reaction.

### 2.5. Scanning and Quantification of Spot Signal Intensities

Chemiluminescent signals were detected on an Odyssey FC (LI-COR Bioscience) using the same conditions described previously [50] with minor modifications. Briefly, a digital image file was generated at a resolution of 5 MP and the signal intensities were quantified using the TotalLab TL100 (v 2009, Nonlinear Dynamics, Newcastle-Upon-Tyne, UK) software. This program has an automatic grid search for 384 spots but does not offer the automatic identification of possible epitope sequences. Due to this, obtained data were analyzed with the aid of a Microsoft Excel program, and to be considered an epitope the sequences of two or more positive contiguous spots should present signal intensity (SI) greater than or equal to 30% of the highest value obtained from the set of spots on the respective membrane. The signal intensity (SI) used as a background was a set of negative controls spotted in each membrane.

### 2.6. Preparation of the Bi-Specific-Antigen Peptides

The single peptide epitopes CB/Tx2 (GSFVMENFSS) and CB/Tx12 (VDIGF) and the peptides CB/Tx4 (KGFYSTDNKY) and CB/Tx13 (SPGHK) were synthesized in tandem containing GG interpeptides as a spacer (Top-Peptide Bio Co., Pudong, Shanghai, China). The peptides were purified by HPLC and their identities were checked by MS (MALDI-TOF or electrospray).

For preparation of the dendrimeric multiantigen peptide (MAP), a standard solid phase synthesis protocol was followed using the tetrameric Fmoc4-Lys2-Lys1-B-Ala Wang resin and the sequence of the biepitope described above. The construct was prepared in an automated peptide synthesizer (PSS8-model, Shimadzu, Kyoto, Japan) as described previously [52].

### 2.7. Enzyme-Linked Immunosorbent Assay (ELISA)

The in-house ELISA was realized as described previously with minor modifications [51]. Briefly, the plates of ELISA were coated with 100 µL (25 µg/mL) of each MAP4 (CB/Tx2–12 and CB/Tx4–13) prepared on coating buffer (Na_2_CO_3_–NaHCO_3_, pH 9.6) overnight at 4 °C. After each incubation step, the plates were washed three times using PBS-T washing buffer (PBS with 0.1% Tween 20 adjusted to pH 7.2) and blocked (200 µL) with 2% defatted milk (2 h at 37 °C). Either the human (huVS) or the mice vaccinated sera (miVS) were dilutes (1:100) in coating buffer and 100 µL was applied onto Immulon 2HB flat-bottom 96-well microtiter plates and incubated for 2 h at 37 °C. Following several washes with PBS-T, the plates were incubated with 100 µL of goat IgG antihuman IgG (1:5000) or rabbit IgG antimice IgG conjugated to biotin (1:2000) for 2 h followed by HRP labeled neutravidin (1:8,000 for human sera and 1: 5000 for mice sera). The plates were developed with TMB (3,3′,5, 5′tetramethyl benzidine) as substrate. The absorbance values at 405 nm were read using an ELX800 Microplate Reader (Bio-Tec Instruments Inc. Winooski, VT, USA.) and the immune response was defined as significantly elevated when the reactivity index (RI; O.D.450 value of target-O.D.450 values of cut-off)) was more than 0.05 M.

Analysis using the commercial diphtheria IgG-ELISA Kit was conducted as described by the manufacturer (Serion Brazil, Paraná, Brazil).

### 2.8. Bioinformatics Tools

The data bank searches for DTx from *C. diphtheriae* (Q6NK15), *C. ulcerans* (A0A1Y0HBB0), and *C. pseudotuberculosis* (WP_014654963.1) were carried out on the database UniProtKB http://www.uniprot.org/ (accessed on 3 March 2020) and NCBI. The alignment of the sequences was performed using the T-Coffee server (HTTP://tcoffee.vital-it.ch/cgi-bin/Tcoffee/tcoffeecgi/index.cgi (accessed on 11 September 2020).

The prediction of the secondary structure of the protein was performed by the PSIPRED servers http://bioinf.cs.ucl.ac.uk/psipred; (accessed on 15 Novembe 2020) and CDM http://gor.bb.iastate.edu/cdm/ (accessed on 21 August 2020). Some physicochemical parameters related to the peptides used in this study (molecular weight, theoretical pI, and hydrophobicity) were calculated with the ProtParam tool http://web.expasy.org/program/ (accessed on 5 August 2020). The prediction of transmembrane domains was obtained using TopCons (https://topcons.net/pred (accessed on 5 January 2021)).

### 2.9. Structural Localization of the IgG Epitopes

The orientation of epitopes in the protein crystallographic structure of DTx protein (PDB: 1xdt) was performed using PyMOL, Molecular Graphics System, Version 2.0 Schrödinger, LLC. For the amino acid sequence of epitopes and transmembrane topology visualization, the platform Protter was used (http://wlab.ethz.ch/protter/ (accessed on 12 December 2020).

### 2.10. Statistical Analysis

Statistical analysis was performed using Graph Pad Prism version 5.0. The statistical difference using a t-test was considered if *p*-value ≤ 0.05.

## 3. Results

### 3.1. Identification of the Immunodominant IgG Epitopes in Diphtheria Toxin

The epitopes in the DTx (560 aa) were identified based on recognition of peptides in a synthesized library by mice antibodies immunized with DTx (see Materials and Methods). In Figure 1, panels A and B present the position of each peptide and the measured intensity, respectively, from the chemiluminescent detection of mice IgG antibodies in sera pooled from vaccinated mice. The intensities were normalized using 100% as defined by the positive control (data not shown). The list of the synthetic peptides and their positions on the membranes is presented in Appendix A.

The pattern of reactivity for the antibodies generated in mice immunized with DTx demonstrated that a greater number of peptides were recognized (Figure 1). An analysis of the sequences constituting the peptides synthesized in reactive regions defined 20 IgG epitopes generated by the DTx vaccine (Table 1).

### 3.2. Identification of the IgG Epitopes in SF23 Peptide Sequence

The SF23 peptide sequence (aa533–aa555; SIGVLGYQKTVDHTKVNSKLSLF) corresponds to the receptor bind fragment of DTx and forms a beta harping structure [53], which binds to the cellular receptor HBGF. The antibodies are capable of blocking the infection [54,55]. Thus, in this assay, a library of 15 peptides covering sequences aa518–aa547 was produced (Figure 2) and probed with sera of huVS, miVS, and hoTS. The amino acids in the strand involved in the β harping structure (aa535 to aa541) do not feature in the epitope structure. The results are presented in Figure 2 and Appendix A shows that this segment displays the major blocking antigenic sequence recognized by either miVS (Figure 2A; KIH*SNEISSDSI*; aa523–aa534), huVS (Figure 2B; RSSSEKIH***S**NEILSDSI*; aa517–aa534), or hoThe sera (Figure 2C; RSSSEKIH*SNEISSDSI*, aa518–aa534). The identified sequence appears to be only part of the described SF23 peptide sequence and contains the CB/Tx-20 epitope (Table 1). Some additional amino acids positioned at the N-terminus of the CB/Tx-20 epitope have been recognized by IgG antibodies depending on the species (Appendix A).

### 3.3. Cross-Immune IgG Epitopes

To investigate the cross-immunity conferred by the DTx protein, the *C. diphtheria* (Q6NK15), *C. ulcerans* (AQA1Y0HBB0, 97.5% similarity) and *C. pseudotuberculosis* (WP_014654963.1, 93.93% similarity) deposited in NCBI data bank were aligned to compare the epitope sequences. The results shown that 100% of the epitopes were common, indicating that the immunity is genus-specific (Appendix A). In other analysis, the epitopes of the DTx protein were compared to the putative toxin sequences of *C. botulinum* and *C. tetani*, but in this case no cross-reactivity was detected (data not shown).

### 3.4. Epitope Reactivity by ELISA

The serological cross-reactivity of bacterial proteins is well known. The impetus for identifying the linear B-cell epitopes in DTx was to improve their use as antigens for diagnostic ELISA assays. To prove their utility, two biepitope peptides were synthesized from sequences derived from the potential CB/DTx-2/miG (SFVMENFSSY) and CB/DTx-12/miG (VDIGF) epitopes and CB/DTx-4/miG (KGFYSTDNKY) and CB/DTx-13/miG (SPGHK) (Table 1), from the major antigenic regions in DTx by solid phase synthesis using the F-moc strategy. The two 17 mer peptides were synthesized as single peptides as four branch MAPs (DTx2/12-MAP4 and DTx4/13-MAP4) contain the dipeptide GG interpeptides purified by HPLC gel filtration and analyzed by MASS spectrometry.

To verify the diagnostic performance of the peptides, sera from ninety-two DTP vaccinated individuals were tested on an ELISA-peptide based assay (Figure 3A,B). All 92 DTP-positive human sera and 38 mice sera reacted with the highly conserved DTx2–12 and DTx4–13 peptides but did not react with peptide EP7 [56] from *T. cruzi* (PPGEDMHTRDGPRE), considered by a scale value as a good test. Both CB/DTx2–12 and CB/DTx4–13 antigens demonstrated 100 and 99.96% sensitivity, respectively, and 100% specificity (Figure 3C,D). Similar results were obtained using either single or MAP4.

### 3.5. The Spatial Location of Enterotoxins A, B, and P IgG Reactive Epitopes

To analyze the localization of epitopes in DTx, the crystallographic structure [57] available in PDB (PDB: 1xdt) was used (Figure 4), which displays the spatial location of the 20 reactive epitopes identified by the SPOT synthesis array experiments (Table 1). Fourteen of the identified epitopes were exclusively in loop/coil (CB/DTx1, CB/DTx3–12, CB/DTx14, CB/DTx16, and CB/DTx20) structures while three (CB/Tx-2, XCB/Tx13, CB/Tx15) shared amino acids in coil and helix, and three (CB/Tx17–19) in coil and strain (Table 1 and Figure 4). However, all were present on the protein surface and accessible to the solvent (data not shown).

In Figure 5 is presented a linear model showing the single transmembrane domains of the protein and the intra- and extracellular distribution of the epitopes.

## 4. Discussion

The high-throughput immune-profiling peptide array formed directly onto cellulose membranes allowed the identification of the major antigenic determinants of the DTx recognized by sera of mice orally vaccinated with a single dose of the DTP. Twenty epitopes covering the full extent of the bacterial protein were identified (Table 1). This toxin is secreted by corynebacterium and causes disease in humans by inhibiting protein synthesis. It possesses a molecular weight of 62 kDa and can exist as a monomer (when secreted) or dimer, with two subunities linked by SH bridges. Three well-defined domains were identified based on trypsin hydrolysis [16,17]. Two epitopes (CB/DTx-1–2) were positioned in the signal peptide (aa1—aa38), seven epitopes (CB/DTx2–8) in the catalytic domain (CD, aa39–aa188), six in the pore-forming membrane-translocation (PFMT) domain (CB/DTx-10–15, aa200–aa378), and five in the receptor binding domain (RBD, aa387–aa535) (CB/DTx-16–20).

The spatial folding of proteins plays a decisive role in determining their antigenic specificity, and it has been useful to distinguish between sequential and conformational determinants. The current production process for DTx vaccines is largely based on the traditional methods first used to detoxify DTx [58]. Recently, Metzel et al. [59] reveled the chemical modification of the DTx induced by formaldehyde treatment. The detoxification used for vaccine preparation typically resulted in a combination of intramolecular cross-links and formaldehyde-glycine attachments. In this process, both the NADþ-binding cavity and the receptor-binding site of diphtheria toxin are chemically modified. Functionally the process affected CD4þ T-cell epitopes to some extent but universal CD4þ T-cell epitopes remained almost completely unaltered [59]. In our study, we can conclude that apparently the detoxification process does not interfere in the recognition of linear B epitopes, since antibodies present in the sera of mice immunized recognized almost of all linear epitopes of the DTx.

In a synthetic model, we have shown that all the twenty B-linear epitopes identified are positioned on the surface of the molecule, and thus are exposed to the immune system. Thus, based on the function of the three domains, apparently the 20 identified epitopes can present important functions such as the blocking of the enzymatic activity of the DTx (CB/DTx2–8), or intracellular translocation (CB/DTx-10–15, aa200–aa378) and sterical impairment of DTx receptor binding (CB/Tx-16–20). To our knowledge, there are only a few studies concerning the identification of DTx key epitopes responsible for vaccine protection. Studies conducted by others produced antibodies in guinea pigs against a synthetic peptide covering the sequence aa186–aa201 [53]. This epitope corresponds to the epitope CB/DTx-9 (aa196GKRGQ200). Another study established that humAb binds to the RBD and physically blocks the toxin from binding to the putative receptor, heparin-binding epidermal growth factor-like growth factor [52]. Murine mAb capable of binding to A and B subunits and neutralizing the DTx-mediated cytotoxicity to Vero cells has also been obtained [60], but in these last two cases, no epitope identification was conducted.

Another important motif of the DTx studied is the RBD. A beta-hairpin synthetic peptide (SF23, aa526–aa548; SIGVLGYQRTVDHTKVNSKLSLF) has been considered as a promising candidate for the development of a synthetic vaccine [52]. This peptide binds to antibodies from the sera of persons infected by toxigenic *C. diphtheria,* those immunized by diphtheria toxoid, and hoThe sera, and block the interactions with the HBEGF cellular receptor [52]. This beta-hairpin sequence encompasses only part of the CB/DTx-20 epitope (aa526–aa540) identified by us using immunized mice and children sera (Table 1). Due to the biological importance of this peptide sequence, a new round of Spot synthesis analysis was performed (Figure 3). A small library containing mutated amino acids was synthesized and reacted with both human and mice vaccine sera and hoThe sera. This study demonstrated that a central peptide amino acid sequence is recognized by sera from miVc, huVac, and hoThe, but small N and C termini aa variation recognition occurred depending on the vaccinated species (Figure 1B).

In all clinical cases, the primary therapeutic option is still treated with equine hyperimmune sera anti-DTx. However, the production of therapeutic antibodies in horses raises ethical issues surrounding the use of animals, especially by substandard housing and veterinary care of the horses. Additionally, hoThe is of limited supply and often unavailable to patients due to interrupted production in several countries [61]. Hence, it is important to find alternatives to replace the equine sera anti-DTx.

One of these approaches has been the use of phage display to obtain neutralizing recombinant human antibodies [45]. However, the success of this methodology seems low since repetitive epitopes with different sizes and longer sequences, encompassing at least six of our identified epitopes, have been recently described [45]. On the other hand, a potent toxin neutralization would require blockage of several points or domains of the molecular surface, which could not be covered by a single antibody. Therefore, our epitope mapping on the planar surface provides superior information describing with greater coverage the set of IgG epitopes responsible for the neutralization of DTx.

The epitopes identified in this study can serve as a basis for the construction of a recombinant polyprotein that can replace the DTx in vaccine preparations since they were identified for antibodies from human, mice, and hoThe sera. The improvement of the vaccine would allow the detoxification stage to be avoided, a fact that causes side effects, as well as introduce cost reduction and greater efficiency to the process [58].

Another interesting finding in our study was the identification of two epitopes (CB/DTx 1–2) present in the signal peptide using serum from mice vaccinated with the DTP vaccine. The presence of human antibodies to one of these epitopes was also hardly detected by ELISA using sera from vaccinated infants (Figure 4). This fact means that a reasonable amount of diphtheria toxoid used in vaccination processes contains the signal peptide since it is able to stimulate the immune system. Although this fact is important in terms of quality control of the production of immunizers, in terms of vaccination it should not represent any counterpoint or impediment. However, it should be noted that a significant amount of proteins did not mature during cell processing of vaccine manufacture.

Diphtheria is caused by toxin-producing corynebacteria. Not only *C. diphtheriae*, but also *C. ulcerans* and *C. pseudotuberculosis* have the potential to produce DTx and hence can cause classic respiratory diphtheria as well as cutaneous diphtheria [11]. It is noteworthy that, in recent years, recrudescence of *C. ulcerans* infections has been reported worldwide and fatal infections have occurred [25,28]. *C. diphtheriae* strains, whether toxigenic or nontoxigenic, have been found to also cause severe and often fatal systemic infections in immunized subjects [25,26,27,62]. DTx produced by the different corynebacteria (*C. diphtheriae,*
*C. ulcerans*, and *C. pseudotuberculosis)* differ at the nucleotide and amino acid levels but are immunologically identical, as shown in this work (Appendix A). Other similar toxins from *C. tetani* and *C. botulinum* were also analyzed and although some cross-reactivity against the SF23 peptide has been described [52], in our study no common epitope such as those described for DTx was identified (data not shown).

Concerning the diagnosis, the ELISA results using the MAP4 biepitope peptides revealed that all epitopes appropriately discriminated between negative and positive samples (*p* < 0.0001). No cross-reactivity was observed, as anticipated from the BLAST analysis criteria. In comparison to the commercial kit, the use of the biepitope improved the performance of our in-house ELISA by generating a higher signal that showed a greater difference between positive and negative samples. From the ROC analysis, the sensitivity of the in-house ELISA peptides was 99.96–100%, similar to the commercial kit. This increase in signal most likely reflects the performance of the selected epitopes. Another benefit of the biepitope protein design was to increase the specific activity of the molecule to bind antibodies.

Phase IIA comprises studies designed to estimate the accuracy (sensitivity and specificity) of the index test in discriminating between diseased and nondiseased people in a clinically relevant population [63]. For diphtheria, the scope of the likelihood score of the ROC analysis was obtained since a much larger and diverse panel of negative controls was available to support our conclusion that the MAPs have 100% specificity. The performance of selected epitopes strongly supports continued use of these epitopes with a chimeric multiepitope protein [64] as a target in more sensitive and fast diagnostic tests and suggests that the antigen may be eligible to enter phase IIB studies [62].

## 5. Conclusions

In this work, we have provided new insights describing, with greater coverage, the set of 20 IgG linear B epitopes responsible for the neutralization of DTx and demonstrated the specificity and eligibility to enter phase IIB studies of some epitopes to develop new and fast diagnostic assays. Additionally, they can serve as a basis for the production of a recombinant vaccine carrying these epitopes, which would avoid the need for a detoxification process which would also reduce production costs.

## Figures and Tables

**Figure 1 vaccines-09-00313-f001:**
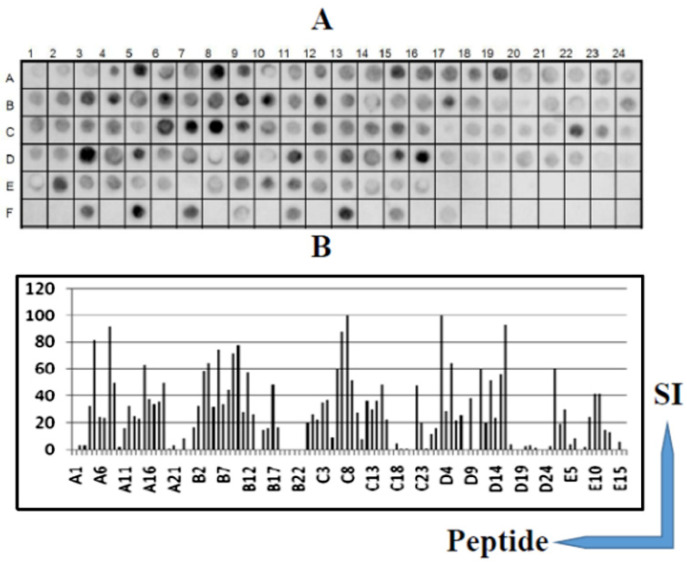
B-cell linear epitopes mapping along the primary sequence (560 residues, 112 peptides) of the diphtheria toxin (DTx, Q6NK15) protein in the SPOT synthesis array (15 residues with overlapping of 10) with the sera (*n* = 15) of DTP vaccinated mice. IgG-reactive peptides (**A**); signal intensity (SI) (**B**). The peptides (listed in Appendix A) represent the coding region of DTx protein. Each peptide was identified by the SPOT synthesis membrane position numbering. Spot intensities below 20% were considered as background.

**Figure 2 vaccines-09-00313-f002:**
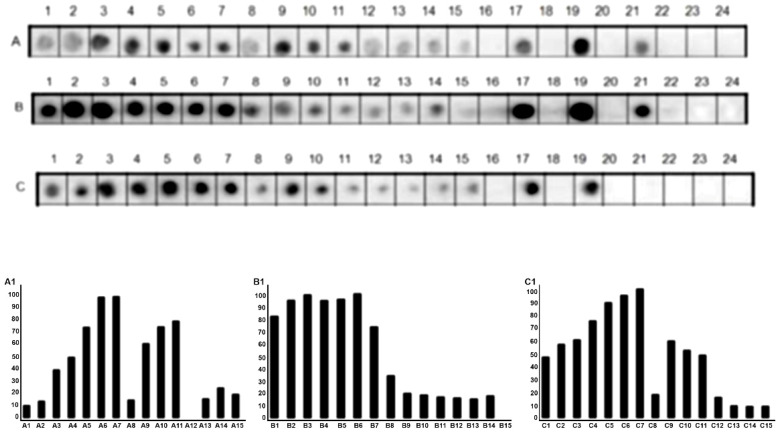
Reanalysis of the DTx sequence (aa518–aa547) by SPOT synthesis (15 aa with overlapping of 14 residues) probed with miVc ((**A**), 1:150), human vaccinated sera (huVS) ((**B**), 1:150) and hoThe sera ((**C**), 1:200). Below, measurement of the signal intensity (A1, A2, A3). Spots 17, 19, 21 are positive controls, respectively (KEVPALTAVETGATN (Poliovirus), GYPKDGNAFNNDRI (*Clostridium tetani*) and YDYDVPDYAGYP YDV (*H. influenza* virus hemagglutinin)). Spots 18, 20, 23, 24 (without peptides). The overlapping of positive spots is presented in Appendix A.

**Figure 3 vaccines-09-00313-f003:**
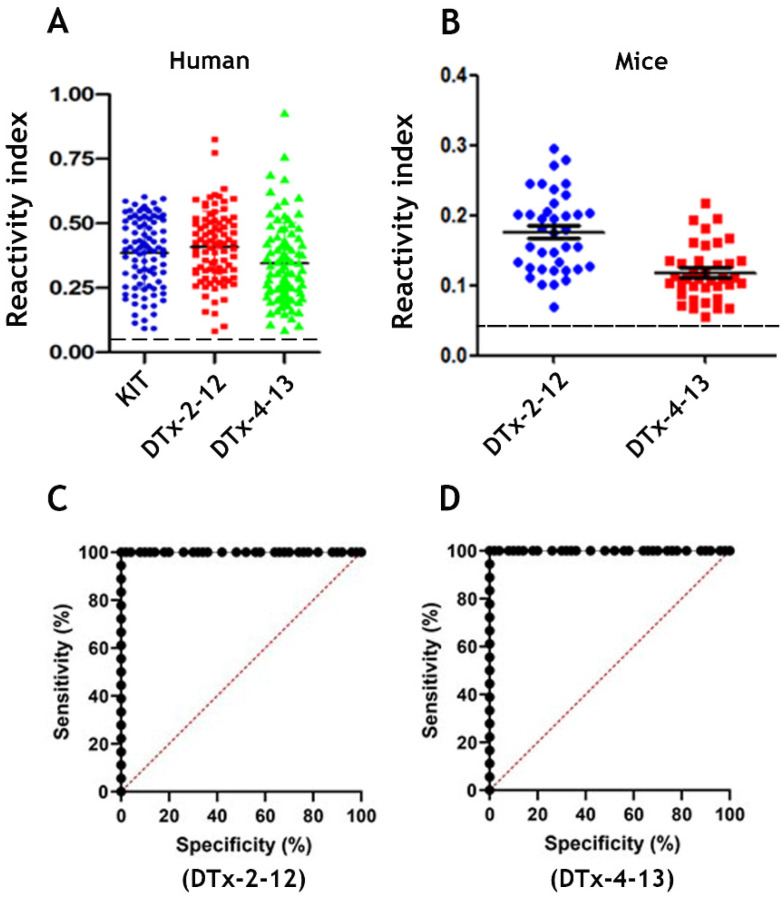
Reactivity of huVS ((**A**), *n* = 90), and mice vaccinated sera (miVS) ((**B**), *n* = 38) with synthetic bispecific spanning epitopes MAP4 in an ELISA compared with a commercial kit ((**A**), *n* = 92). The *y*-axis shows the mean reactivity (positive/mean negative) of sera from vaccinated patients and mice. The ROC analysis (**C**,**D**) using huVS showed that the sensitivity of the in-house ELISA peptide was 100% for DTx-2–12 and 99.96% for DTx4–13 in a range similar to the commercial kit, with a specificity of 100%. The dashed lines in (**A**,**B**) indicates the cut-off values of the tests.

**Figure 4 vaccines-09-00313-f004:**
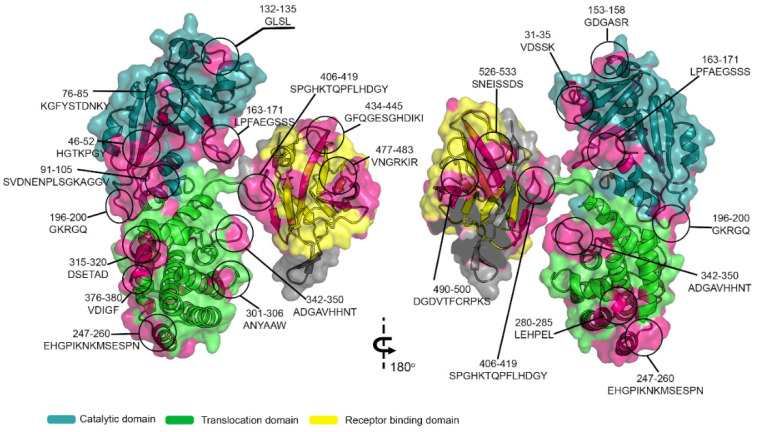
Three-dimensional structure of DTx (Q6NK15) with the signal peptide and position of the epitopes identified by SPOT synthesis. Molecular model overlay of neutralizing epitopes within the DTx protein. Image constructed using crystal structure of toxin (PDB: 1xdt), showing catalytic (**cyan**), translocation (**green**) and receptor binding (**yellow**) domains. Identified epitope sequences were highlighted in magenta. Image was created using PyMOL.

**Figure 5 vaccines-09-00313-f005:**
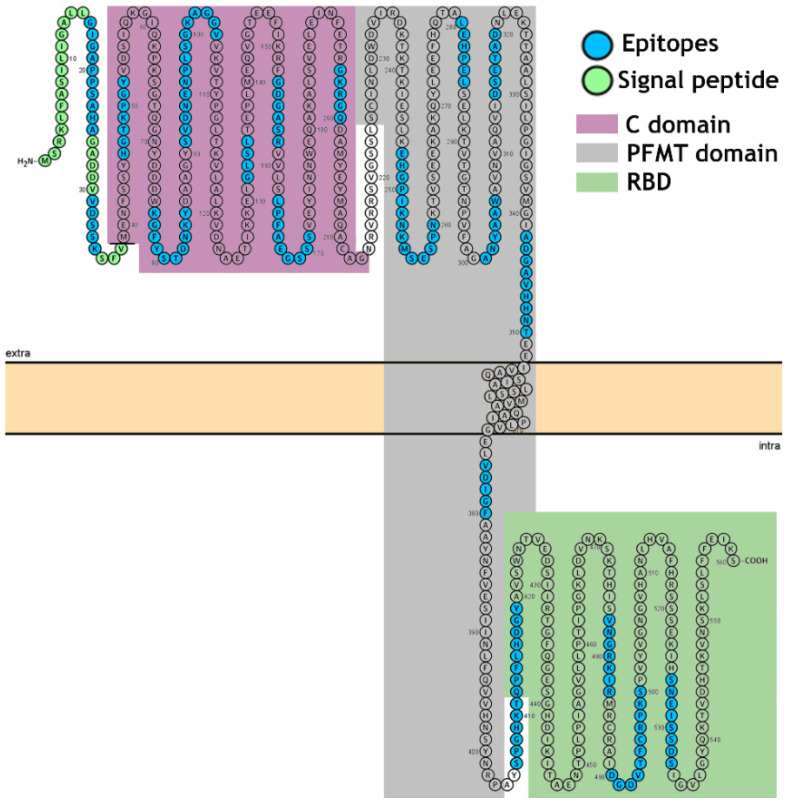
Modeling of DTx (Q6NK15) of *C. diphtheria* as a single-pass transmembrane protein. The model is based on the predictive results obtained using TopCons https://topcons.net/pred (accessed on 9 January 2021), and layout was generated using Protter http://wlab.ethz.ch/protter/ (accessed on 8 January 2021). The model with the TM (aa353–aa373) in the pore-forming membrane domain (PFMT) reveals all 20 identified epitopes; 1–38, signal peptide; aa39–aa188, C domain; aa200–aa378, PFMT domain and aa-387–aa535, receptor-binding domain (RBD).

**Table 1 vaccines-09-00313-t001:** List of the identified CB/TxD IgG epitopes and secondary structure (C, coil; H, helix; S, strand) based on I-TASSER prediction (https://zhanglab.ccmb.med.umich.edu/I-TASSER/ (accessed on 15 October 2020).

Name of Epitope	Sequence	Secondary Structure	Name of Epitope	Sequence	Secondary Structure
CB/DTX-1	^16^GIGAPPSAHA^25^	C	CB/DTx-11	^280^LEHPEL^285^	C
CB/DTx-2	^31^VDSSK^35^	C+H	CB/DTx-12	^301^ANYAAW^306^	C
CB/DTx-3	^46^HGTKPGY^52^	C	CB/DTx-13	^315^DSETAD^320^	C+H
CB/DTx-4	^76^KGFYSTDNKY^85^	C	CB/DTx-14	^342^ADGAVHHNT^350^	C
CB/DTx-5	^91^SVDNENPLSGKAGGV^105^	C	CB/DTx-15	^376^VDIGF^380^	C+H
CB/DTx-6	^132^GLSL^135^	C	CB/DTx-16	^406^SPGHKTQPFLHDGY^419^	C
CB/DTx-7	^153^GDGASR^158^	C	CB/DTX-17	^434^GFQGESGHDIKI^445^	C+S
CB/DTx-8	^163^LPFAEGSSS^171^	C	CB/DTx-18	^477^VNGRKIR^483^	C+S
CB/DTx-9	^196^GKRGQ^200^	C	CB/DTx-19	^490^DGDVTFCRPKS^500^	C+S
CB/DTx-10	^247^EHGPIKNKMSESPN^260^	C	CB/DTx-20	^526^ SNEISSDS^533^	C

## Data Availability

The data presented in this study are available on request from the corresponding author.

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
