# Peer review of "Epitope Mapping of the Diphtheria Toxin and Development of an ELISA-Specific Diagnostic Assay"

_vaccines, 2021, doi:10.3390/vaccines9040313_

Round 1

Reviewer 1 Report

This research article by De-Simone S. et al. provided detailed epitope maps in the diphtheria toxoid antigen by the Spot-synthesis membrane position numbering using human, mouse and horse serum. The experiments confirmed and extended the previous reports that included identification of new epitopes in the antigen. The findings were discussed well. To conclude, the article provides meaningful information and opinions on this field concisely and timely and it fits well to the scope of this journal. I will recommend the publication of the manuscript with a minor modification. Minor points to consider in subsequent versions are shown below.

 Page 3 Line 15. The commercial horse therapeutic (hoThe) sera was obtained from Butantan Insti-tute, São Paulo, SP, Brazil. The detailed information of this reagent should be provided.

 Figure 2. The information for I and II is too short. The authors should provide more information to avoid confusion. I believe that (I) indicates Spot membrane and (II) indicates measurement of the signal intensity. There are 24 spots, but data for spots have only 15 lines. The resolution for the figure is very low to check numbers.

 Table S1 showed sequence in F3, F5, F7, F9, F11, F13. F15, F7, but the legend mentioned “List of synthesized peptides covering the entire sequence of DTx. (F3,F4,F5, F11,F12,F13 positive controls and F9,F17 negative controls”. This difference needs to be fixed.

Author Response

1) Page 3 Line 15. The commercial hoThe… The detailed information … should be provided.

R-The figure caption has been modified, making the information clearer.

2) Figure 2. The information for I and II is too short. The authors should provide more information to avoid confusion. I believe that (I) indicates Spot membrane and (II) indicates measurement of the signal intensity. There are 24 spots, but data for spots have only 15 lines. The resolution for the figure is very low to check numbers.

R-The figure caption has been modified, making the information clearer.

3)3) Table S1 showed sequence in F3, F5, F7, F9, F11, F13. F15, F7, but the legend mentioned “List of synthesized peptides covering the entire sequence of DTx. (F3, F4, F5, F11, F12, F13 positive controls and F9,F17 negative controls”. This difference needs to be fixed.

R- Thank you, this was corrected

Reviewer 2 Report

De Simone et al, presents in the manuscript Epitope mapping of the diphtheria toxin and development of an ELISA-specific diagnostic assay an important aspect to ease the diagnosis of a disease with the potential of becoming a world threat.

The subject is very interesting and sets the base for a diagnosis method and with the possibility to continue a study for vaccine development.

The study's protocol is well designed, with valuable results and significant conclusions.

I would only have a few comments for the authors:

  • please be careful to verify the specie names - in some cases there are misspelled  (I couldn't indicate the exact place, since the line numbers in the manuscript are missing)
  • please shorten the introduction and make it more specific
  • please improve the conclusion section. from the detailed results you could extract more conclusions.

Author Response

  • Please be careful to verify the specie names - in some cases there are misspelled (I couldn't indicate the exact place, since the line numbers in the manuscript are missing)
  • Please shorten the introduction and make it more specific please improve the conclusion section from the detailed results you could extract more conclusions.

R- A search was performed by the automatic word corrector and the names of the correct species. The conclusion was improved

Reviewer 3 Report

The submitted manuscript, De-Simone et al, is an interesting study to define antigen on diphtheria toxin. In this manuscript, the authors have evaluated 112 15-mer peptides that cover entire protein sequence of DTx protein to identify continuous B-cell epitopes and identified 12 peptides that could detect antibodies in huVS and miVS. The manuscript is well written and describe all sections in explicit details. This paper represents an important contribution to vaccine community and will be useful for the broad audience of MDPI vaccines.

My comments:

  1. The authors claim to present novel approach to identify epitopes on diphtheria toxin. However, similar studies have been performed to map epitopes using different techniques. For example, check PMIDs: 30560647. Please comment and compare your epitope mapping result with previously published findings.
  2. Manuscript is full of incorrect sentences. Please correct entire manuscript for English. For example, please correct following sentences in the introduction:
    1. “…….Is endemic in Asia…..”
    2. “….. in the last few years is has been re-emergent…”
    3. “…… The diphtheria toxoid antigen is a major component in pediatric and booster combi-nation vaccines is known to raise a protective……”
    4. “….however, in most of the epitopes were not identified or longer peptides sequences have been proposed…”
    5. “….DTx hoThe antibodies are capable to block the infection….” What is DTxhoThe?
    6. “…..is only part of the CB/Tx-20/miG epitope…” It should be CB/DTx-20/mIg rather CB/Tx-20/miG.
  3. In current study, the authors performed a peptide-based study and thus the approach covers only linear epitopes in the antigen. Please comment about the role of non-linear (conformational) epitopes on DTx.

Other important comments:

  1. Section 2.3:
    1. How did the authors validate the synthesized peptides for correct sequences?
    2. Please indicate the final number of SPOT membranes and # of peptides/membrane
  2. Section 2.4: Please define SPOT
  3. Section 3.4: “… sera from ninety-two individual DTP vaccined were tested…” As per the method, only 90 vaccinated individuals were used in the study. Please comment.

Figures and Tables:

  1. Figure 1: Please include DPx sequence numbering as the additional X-axis to simplify the peptide location on DPx.
  2. Figure 2: SF23 sequence, SIGVLGYQKTVDHTKVNSKLSLF, is not present in peptides in panel III. Please comment.
  3. Figure 3 C and D: what does the significance of diagonal red-dotted line shown in the figures?
  4. Figure 5:
    1. Please provide uniprot ID of DTx of diphtheria in the figure legend.
    2. The shown model protein crosses cell membrane however no predicted transmembrane is shown. Please comment.
    3. The model shows two epitopes localized to signal peptide region. Please comment on the significance of such epitopes in signal peptide region and provide relevant publication to support their inference.

Minor comments:

Introduction

  1. …..DTxs produced by the different corynebacteria are structurally very similar…”

What does very similar means? Please provide sequence identity or rmsd values for aligned structures.

Result

  1. Section 3.5:
    1. “To analyze the localization of epitopes in DTx, the crystallographic structure available in PDB (PDB: 1xdt) was…”

Please cite the relevant publication for PDB ID 1xdt.

  1. “..Most of the identified epitopes were…”

Rather stating most identified, please mention result as # epitopes in coil, loop region/total # epitopes

Author Response

Referee 3

1.    The authors claim to present novel approach to identify epitopes on diphtheria toxin. However, similar studies have been performed to map epitopes using different techniques. For example, check PMIDs: 30560647. Please comment and compare your epitope mapping result with previously published findings.

The MS: 30560647 (Asian Pac J Allergy Immunol 2008; 26: 47-55), was checked and describes obtaining mAb that bind to different sites of A and B protein and block the enzyme activity of the toxin, at no time maps epitopes. Thus, it is difficult to discuss and compare these results due to the lack of information about the epitopes involved. Regardless, a sentence describing the experiments was introduced (please see line 340-342)[60].

As far as we were able to search in PubMed and other indexers, all references describing DTx “epitopes” were cited in the manuscript, especially in the introduction. As they have used different techniques, and mostly describing epitopes of long stretches, I consider it out of purpose to discuss the subject.

2)      For example, please correct following sentences in the introduction:

“…….Is endemic in Asia…..”

“….. in the last few years is has been re-emergent…”

“…… The diphtheria toxoid antigen is a major component in pediatric and booster combi-nation vaccines is known to raise a protective……”

“….however, in most of the epitopes were not identified or longer peptides sequences have been proposed…”

“….DTx hoThe antibodies are capable to block the infection….” What is DTxhoThe?

“…..is only part of the CB/Tx-20/miG epitope…” It should be CB/DTx-20/mIg rather CB/Tx-20/miG.

Thank you, all notation were corrected in the text. Please see lines 41, 71, 80, 237 and 242.

3) Manuscript is full of incorrect sentences. Please correct entire manuscript for English.

A native-language proofreader corrected the text.

4) In current study, the authors performed a peptide-based study and thus the approach covers only linear epitopes in the antigen. Please comment about the role of non-linear (conformational) epitopes on DTx.

The few descriptions of conformational epitopes were mentioned in the introduction (see lines 75,76). However, as the toxin is used in the vaccine after detoxification with formalin (ie denatured), the conformational epitopes lose their protective importance. Therefore, we have not identified any importance in discussing this point of view.

5) Other important comments:

1.    Section 2.3: -How did the authors validate the synthesized peptides for correct sequences?- Please indicate the final number of SPOT membranes and # of peptides/membrane

2.    Section 2.4: Please define SPOT

3.    Section 3.4: “… sera from ninety-two individual DTP vaccined were tested…” As per the method, only 90 vaccinated individuals were used in the study. Please comment.

4.    Figures and Tables:

a-Figure 1: Please include DPx sequence numbering as the additional X-axis to simplify the peptide location on DPx.

b-Figure 2: SF23 sequence, SIGVLGYQKTVDHTKVNSKLSLF, is not present in peptides in panel III. Please comment.

c-Figure 3 C and D: what does the significance of diagonal red-dotted line shown in the figures?

d-Figure 5: -Please provide uniprot ID of DTx of diphtheria in the figure legend.

e- The shown model protein crosses cell membrane however no predicted transmembrane is shown. Please comment.

f-The model shows two epitopes localized to signal peptide region. Please comment on the significance of such epitopes in signal peptide region and provide relevant publication to support their inference.

1-When a peptide library is synthesized by the Spot Synthesis technique or parallel synthesis there is no necessity of external "validation". In other word, the validation is intrinsic since that the peptides are synthesized in parallel (for overlapping) immobilized covalently in the cellulosic membrane. The control of the entry of each aa in each cycle is done by the computer program in which the complete sequence taken directly from the database is made available. The validation necessity of peptide sequences is done only when the synthesis occurs in "solution".

2-We do not think that this information is necessary to understand the paper. The “SPOT” technique existed since 1994 and is consecrated in the world literature of peptide chemistry. However, to clarify the referee, the name SPOT is already self-defining, and in peptide chemistry it means the synthesis of peptides in an imaginary concentric halo where the “point” on the membrane functions as a reaction chamber.

3- The numbers of sera have been corrected in the text

4a-We do not think it necessary to modify the X-axis of the Fig 1. Peptide defines the number of synthesized peptide and if we change it to DPx it can be confused with the code given to the DTx epitopes. In addition, in the list of synthesized peptides exist controls non-DTx (see supplementary Table S1).

4b- Thanks for the observations, figure 2C really had conflicting data and the figure blurred. So the data was corrected and a new figure was made. However, as it became very large, we found it better to insert Figure 2C as a supplementary Figure S1.

4c-It means the cut-off of reactions. This information was included in the legend of figure 3.

4d-This information has been included.

4e-The sequence shown in the transmembrane region of the figure, already informs the TM domain. However, to make it clearer as suggested, the aa of the TM sequence were described in the figure legend.

4f-Through my search we were unable to identify any work that comments on the immunological or vaccine importance of the signal peptide. The entire signal peptide is removed from the nascent protein during synthesis. Our description is the first to raise the issue, and proves that a certain amount of "mature" protein is present in vaccine preparations.
